# Evaluation of the Filming and Protective Properties of a New Trehalose and Ceramides Based Ingredient

**Letteria Greco [1,\*]** , **Salviana Ullo [2,\*]**, **Luigi Rigano [3]**, **Marco Fontana [1]**, **Enzo Berardesca [4]** and **Norma Cameli [4]**

[1]  Sinerga Group, Via della Pacciarna 67, 21050 Gorla Maggiore, VA, Italy; m.fontana@sinerga.it
[2]  Sinerga SPA, Via della Pacciarna 67, 21050 Gorla Maggiore, VA, Italy
[3]  ISPE, Institute of Skin and Product Evaluation, Via Giuseppe Bruschetti, 1, 20125 Milano, MI, Italy; info@ilcosmetologo.com
[4]  Istituto Dermatologico S. Maria e S. Gallicano, IRRCS Via Chianesi 53, 00144 Roma, Italy; berardesca@gmail.com (E.B.); normacameli@gmail.com (N.C.)
**\***  Correspondence: l.greco@sinerga.it (L.G.); s.ullo@sinerga.it (S.U.)

**Abstract:** The aim of this study is showing the filming and skin barrier protective properties of a new ingredient based on ceramides and trehalose and carried in lipophilic vesicles composed of lecithin and cholesterol (or phytosterols). Through an in vivo study, the restructuring and hydrating properties of this trehalose and ceramides compound have been evaluated. Furthermore, this new ingredient has been used in a topical formulation for atopic dermatitis, proving to be effective in the alterations of skin barrier. This evidence makes it an interesting ingredient for topical dermatological compositions in the treatment of dermatitis and all manifestations correlated to these skin disorders, such as edema, swelling, rash, redness, and itching. Its soothing and protective action against the painful and annoying symptoms like those given by dermatitis makes this trehalose and ceramides based ingredient for topical use.

**Keywords:** trehalose; ceramides; lecithin; skin disorders; dermatitis; xerosis

## 1. Introduction

The Stratum Corneum (SC) protects the body from the loss of physiologically important components, as well as against harmful environmental insults. It contains about 15 layers of corneocytes separated by a unique and complex mixture of highly ordered multilamellar lipid sheets, which is often referred to as a brick wall-like structure. Chemically, the whole SC contains about 5–15% lipids, 75–80% proteins, and 5–10% unknown materials on a dry-weight basis. However, it is the small percentage of intercellular lipids in the SC that defines the only continuous, tortuous pathway through which molecules can diffuse across the SC and that plays a major role in selective permeability and skin barrier function [1].

Corneocytes, its main constituents, and the intercellular lamellar lipid bilayers are considered the main structures determining the speed of the transcutaneous exchange of substances. The mechanical resistance of the epidermal barrier is mainly due to the corneocytes embedded in the so-called cornified envelope. The main biochemical components of the skin barrier are proteins, such as the dynamically linked loricrin, involucrin, filaggrin, and lipids, which are responsible for the water permeability and the exchange of substances with the external environment [2].

Dermatological treatments of alterations of skin barrier include lipids, the substances that play a key role in maintaining and restructuring skin barrier.

The object of this article is the study of a new ingredient based on ceramides, trehalose, lecithin, and cholesterol and its efficacy in skin disorders treatment. The blend is composed of a lipophilic matrix in which compounds are incorporated. The solubilization of substances by means of lipophilic matrixes is currently one of the most used techniques in cosmetics and dermo-pharmaceutics, in order to increase the diffusion of active ingredients. "Lipophilic matrix" means a phospholipid- and lecithin derivative-based matrix in combination with appropriate lipophilic carriers such as medium- and long-chain glycerol esters, medium- and long-chain fatty acid esters, mineral and vegetable oils [3].

A new version of this ingredient has been recently developed replacing the cholesterol with phytosterols in order to adhere to a precise willing of not having animals derivatives in the composition of the ingredient. Furthermore, the activity is completely the same, since the main effectivity is brought by ceramides and trehalose, which are included in both versions.

Fifty percent of the lipids involved in the composition of SC are ceramides, which play a crucial role in the organization of the intercellular lipid layers. Double layers of ceramides are covalently bound to proteins of the cornified envelope surrounding the corneocytes, thus determining the homogeneity and unity of the stratum corneum [4–6]. Furthermore, ceramide acts as a water modulator and a permeability barrier by forming multilayered lamellar structures with other lipids between cells in the SC layers. In adult skin affected by atopic dermatitis, there is a ceramide deficiency even in the non-lesional SC, which is highly associated with the abnormal barrier function, predisposing the skin to inflammatory processes evoked by irritants and allergens. The ceramide deficiency in the SC of atopic dermatitis skin has been substantiated also in many additional studies [4].

Regarding trehalose, it is a white, odorless powder with a relative sweetness 45% more than that of sucrose. It is a bisacetal, non-reducing homodisaccharide in which two glucose units are linked together in aa-1,1-glycosidic linkage. It is present quite widely in the biological world and performs different functions. Because of its inherent properties and its stabilization of proteins and lipids, it has proved quite useful in a number of industries, including food processing, cosmetics, and pharmaceutics.

Trehalose has been classified as a kosmotrope or water-structure maker, which is the interaction between trehalose/water is much stronger than water/water interaction and may be involved in its bioprotective action. Because of this property, it is also considered a natural hydrating agent. All its characteristics, both physical and chemical, make trehalose able to stabilize proteins, preserving their aggregation and helping them to maintain their structures during stress induced by dehydration [7–10]. As skin barrier alterations are usually linked to skin dryness conditions, when a humectant sugar agent such as trehalose is associated to this studied ingredient, a substantial increase of the moisturizing rate at the stratum corneum level is observed [3].

The new ingredient is obtained by a specific technique, which implies the production of transparent and/or translucent compositions with particle sizes in a range of nm, obtained through the solubilization in the form of micelles thanks to the use of appropriate emulsifiers and co-emulsifiers and of high-pressure homogenizers and/or microfluidizers [3].

Delivering active substance to the targeted site requires the right concentration of actives in the formulation in order to achieve the optimal release rate and the desired distribution of substances between the vehicle and target site.

Encapsulation techniques are most widely used in the development and production of improved delivery systems [11].

For skin applications, it is possible to use this delivery system in the form of cream or gel. The lipophilic carrier system is used as an absorption promoter so that a better bioavailability and efficacy of previously mentioned components, ceramides and trehalose, are obtained [12]. The size distribution of structures of this new ingredient has been evaluated through transmission electron microscopy (TEM) using the freeze fracturing sample preparation method.

According to performed measurements, the cosmetic ingredient shows content of globular structures with diameter with an average size of 209.14 nm with a standard deviation of 10.69 (Figure 1) [13].

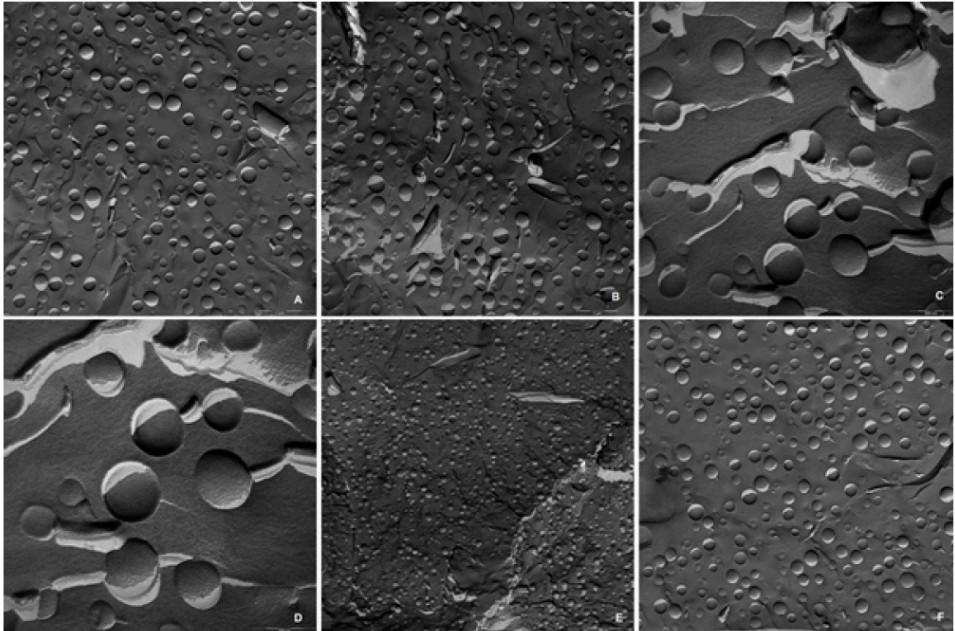

**Figure 1.** TEM Imaging of the analyzed product. Scale: 15,000× (**A,B,F**), 50,000× (**C,D**), 8000× (**E**).

In the light of the high safe profile of the new ingredient, in vivo test has been performed to assess its ability to improve skin barrier effect.

### 1.1. Dermatitis

Dermatitis is a condition characterized by reddened and inflamed skin, irritation, swelling, and itching. The majority of dermatitis has an irritating or allergic origin and corresponds to a reaction to external (allergens, chemicals, physical factors) or internal (release of inflammation elements) features that show up with a widespread rash. They can be temporary or persistent, depending on the causes, and can be complicated by swelling, desquamation, vesicles, blisters, erosions, and crusts.

### 1.2. The Most Common Forms of Dermatitis Are, for Example

**(1) Atopic dermatitis**: very common in childhood, it manifests with redness and blisters at the folds of the skin, e.g., in the elbows, knees, and neck, where moisture promotes skin irritation. Atopic dermatitis is a chronic inflammatory skin disease associated with cutaneous hyper-reactivity to environmental triggers. The clinical phenotype that characterizes this pathology is the product of interactions between susceptibility genes, the environment, defective skin barrier function, and immunologic responses [14].

**(2) Seborrheic dermatitis**: skin inflammation with an intense desquamation, affecting the scalp, central face, and anterior chest. In adolescents and adults, it is often present as scalp scaling (dandruff). Seborrheic dermatitis also may cause mild to marked erythema of the nasolabial fold, often with scales, which are greasy and not dry, as commonly thought. Stress can cause flare-ups. An uncommon generalized form in infants may be linked to immunodeficiencies. Topical therapy primarily consists of antifungal agents and low-potency steroids [15].

**(3) Contact dermatitis**: caused by contact with stinging substances (such as nettle), irritants (such as detergents or other chemicals), or insect poison. It is strongly irritating and can give rise to vesicles in the affected area [16].

### 1.3. In Vivo Efficacy Assessment of a Moisturizer with Skin Barrier Recovering Properties

Atopic dermatitis is a chronic eczema whose main clinical feature is represented by extremely dry skin (xerosis). Such a condition is linked to an impaired skin barrier function.

**Purpose of the test:** the aim of the test is to evaluate the moisturizing efficacy of the ingredient inserted at 2% topical hydrogel, during a four-week period, double-blind, placebo-controlled, on 15 healthy volunteers affected by atopic skin, by noninvasive instrumental measurements.

## 2. Materials and Methods

The assessments were performed by instrumental techniques (Corneometry, Corneometer CM825 Courage & Khazaka, Köln, Germany; Evaporimetry, Tewameter CM210 Courage & Khazaka, Köln, Germany) to measure the amount of water present in the stratum corneum and the skin barrier condition.

Volunteers recruited for this study are 15 healthy subjects aged 18–60 years with very dry skin on limbs.

Exclusion criteria from the study are pregnancy and breastfeeding, allergies, skin lesions affecting measurements on the areas of interest, other medications/cosmetics on the areas under examination within the study period.

The measurements were carried out in basal conditions, after 1 minute from the application of the product and after 3 days, 1 week, 2 weeks, and 4 weeks.

They were taken every time on the same area, selected on limbs, as suggested for each volunteer by reference points scheduled in his/her clinical folder and after a period of acclimatization in a temperature/humidity-controlled environment. Collected data were statistically elaborated by Student T-test for paired (each product vs itself) and unpaired data (1877 vs 1878). Mean ± SD has been considered.

The study was performed through double-blind trial. Each volunteer was given two tubes (one labeled as '1877', the other as '1878') and was instructed to apply one on one side and the second on the other reference area.

Batch n° 1877 corresponds to ceramides and trehalose based ingredient 2% hydrogel.

Batch n° 1878 corresponds to placebo hydrogel.

The volunteers were instructed to use the product twice a day, avoiding applying it at least within 12 h before the assessment. Collected data were statistically elaborated by Student T-test for paired (each product vs itself) and unpaired data (placebo vs ceramides and trehalose based ingredient hydrogel).

## 3. Discussion

The statistical analysis of the collected data at each visit shows a very good trend for hydrogel containing trehalose and ceramides based ingredient at each visit:

**Evaporimetry (Transepidermal Water Loss (TEWL))**

Evaporimetry was used to detect any influence on the water barrier function of the skin following application of ceramides and trehalose based ingredient hydrogel and placebo hydrogel.

**Corneometry (Hydration)**

In order to measure the amount of water present in the stratum corneum and the skin barrier condition, hydration has been evaluated through Corneometry.

Moreover, the T-test applied to hydrogel containing ceramides and trehalose based ingredient and placebo formula gave a statistically significant difference between them at each visit, always showing better results for the hydrogel containing ceramides and trehalose based ingredient.

The double-blind, placebo-controlled study was performed to assess, by corneometry and evaporimetry, the barrier normalizing efficacy of a topical product containing trehalose and ceramides to alleviate skin dryness on limbs.

The study results show a significant moisturizing effect of the ingredient at each visit, both versus baseline and versus the placebo control ($p < 0.01$).

For the application site of topical cosmetic formula containing a trehalose and ceramides based ingredient:

(1) Transepidermal water loss (TEWL) value shows a reduction of −54% after 4 weeks (Figure 2).

(2) Hydration shows an increase of 102% after 4 weeks (Figure 3).

Such results point out the significant moisturizing effect after using the cosmetic under examination and go well with the rationale of the formulation, based on ingredients able to moisturize rapidly the skin and recover the cutaneous barrier to allow it to better hold the water present into its outer layers.

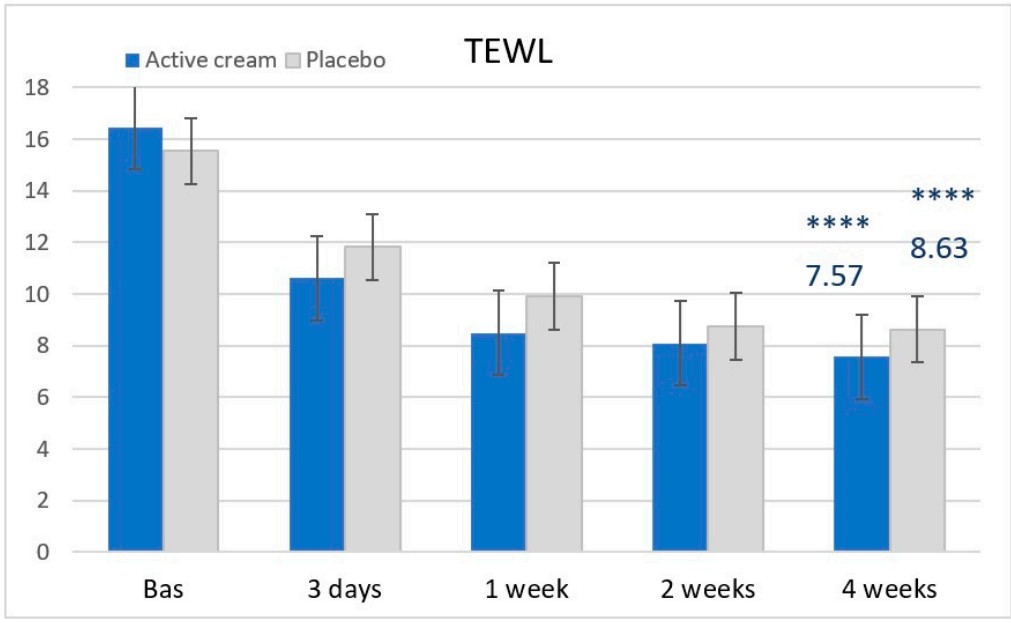

**Figure 2.** Transepidermal water loss (TEWL) values (mean + SD) show a reduction of −54% in the hydrogel (with 2% of ceramides and trehalose based ingredient) site after 4 weeks (**** $p < 0.01$ (*t*-test); $n = 15$ subjects; Bas = Baseline, T0; 4 weeks = after 4 weeks of application).

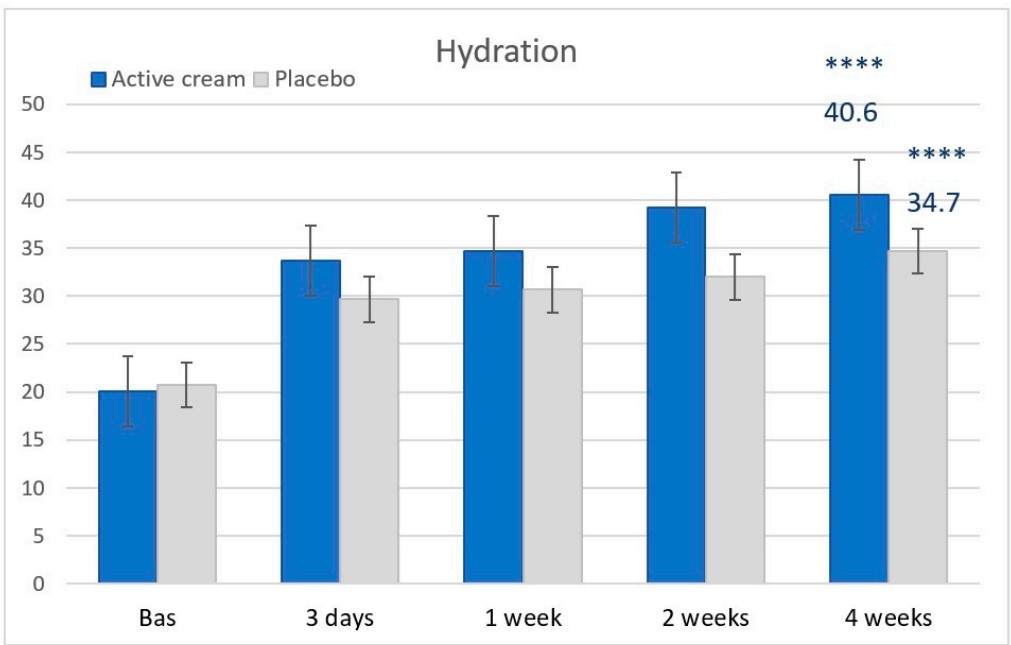

**Figure 3.** Hydration values (mean + SD) show an increase of +102% in the hydrogel with 2% of ceramides and trehalose based ingredient) site after 4 weeks (**** $p < 0.01$ (*t*-test) ; $n = 15$ subjects; Bas = Baseline, T0; 4 weeks = after 4 weeks of application).

in vivo evaluation of the efficacy of a formulation for atopic and very dry skin (Xerosis) was done.

An example of one topical formulation for atopic dermatitis (Table 1) based on trehalose and ceramides based ingredient is reported below for illustrative and non-limiting purposes [17].

**Table 1.** Example of one topical formulation for atopic dermatitis.

| Ingredient | % |
|---|---|
| Water | to 100 |
| Humectant and Functional Ingredients | 0.5–4 |
| Preservative and Chelant System | 0.5–2.5 |
| Rheology Modifier | 0.3–0.8 |
| Emulsifier System | 5–10 |
| Oil Phase | 10–15 |
| Ingredient * | 1–4 |
| pH Adjuster | to ideal pH |

* Trehalose, Ceramides, Lecitihin, Cholesterol/Phytosterols 1% dosage.

Purpose of the test: To provide active substances through a form that allows a prompt release and an effective amount in atopic dermatitis, characterized by a deteriorated functionality of the barrier, an increased transepidermal water loss (TEWL) and a significant xerosis, as well as in the case of pathologies associated with a local subcutaneous microcirculation disorder [3]. The efficacy of a topical cosmetic formulation containing trehalose and ceramides in comparison to a placebo has been evaluated.

Mean values, standard deviations, and variations were calculated for each set of values.

Following the results of normality test (Kolmogorov–Smirnov test), the instrumental data (T0, T4 weeks) and the variations (T4 weeks-T0) were statistically compared by means of *t*-test for parametric and dependent data.

In all cases, the groups of data were considered statistically different for a probability value $p < 0.05$.

Protocol of double-blind study: the product containing trehalose and ceramides ingredient and the placebo was applied on the forearms by 20 volunteers of either sex (1 male and 19 females with an average age of 50 years) with atopic and very dry skin (xerosis), twice a day, for 4 weeks.

During the study, subjects were instructed to wash their body using their current skin care regimen and to not apply the tested products on any other site than the prescribed ones. For the whole duration of the test, the subjects were not allowed to use different products on the forearms and instructed to avoid UV exposure.

Furthermore, the side of application (left or right forearm) of the two formulations (cream containing trehalose and ceramides based ingredient and placebo) were randomized among the volunteers. Each sample was labeled "right" or "left", indicating the side of application of the product. The assignment of subject number and subsequent placement on the randomization chart were made in order of appearance at the study center on the first day. The products were given to the subjects in anonymous containers which did not provide any information about the treatment.

The comparisons evidenced statistically significant differences between studied ingredient and placebo for both parameters (hydration and TEWL).

This means that the improvements observed in the active area were statistically higher than the improvements observed in the placebo area.

Instrumental measurements of skin hydration and transepidermal water loss (TEWL) were performed on a selected area (9 cm$^2$) of the forearms at the baseline and after 4 weeks of treatment [18].

**Skin Hydration**

A statistically significant increase in the mean basal values of skin hydration was evidenced after 4 weeks of application of both the ceramides and trehalose based ingredient cream and the placebo product (Table 2).

**Table 2.** Mean values, standard deviations, variations, and statistical significance (*t*-test) of skin hydration on 20 analyzed subjects.

| Products | T0 | T4 Weeks | Variation (%) T4 Weeks-T0 | P-Level T0 vs. T4 Weeks |
|---|---|---|---|---|
| ceramides and trehalose ingredient based cream | mean 26.9 st. dev. 1.8 | mean 45.6 st. dev. 8.6 | +18.7 (+69.5%) | $p < 0.0001$ |
| placebo cream | mean 26.7 st. dev. 2.2 | mean 42.1 st. dev. 7.7 | +15.4 (+57.7%) | $p < 0.0001$ |

A statistically significant difference between ceramides and trehalose based ingredient cream and placebo product was detected after 4 weeks of treatment (Figure 4).

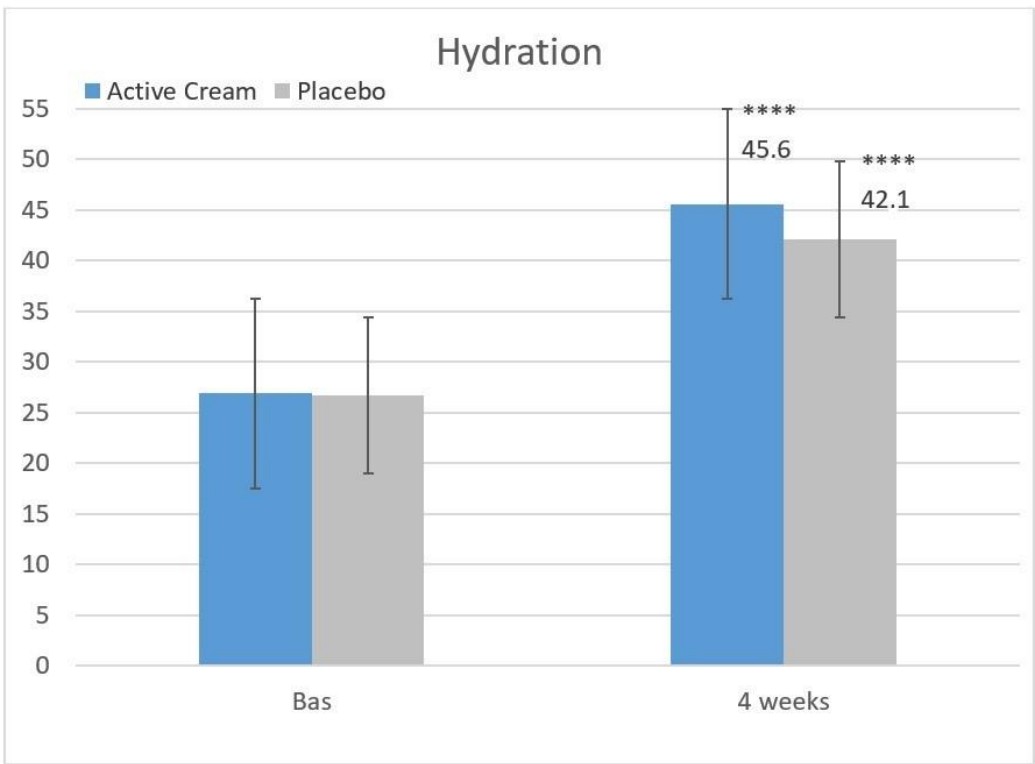

**Figure 4.** Mean values of skin hydration recorded for cream containing studied ingredient and placebo product. Hydration values (mean + SD) show an increase of +69.5% for cream containing 2% trehalose and ceramides based ingredient site after 4 weeks (**** $p < 0.0001$ (*t*-test); $n = 20$ subjects; Bas = Baseline, T0; 4 weeks = after 4 weeks of application).

The cream containing trehalose and ceramides based ingredient is able to improve skin hydration if compared to placebo.

**Transepidermal Water Loss (TEWL)**

A statistically significant decrease (improvement) in the mean basal values of transepidermal water loss was recorded after 4 weeks of application of both the Trehalose and ceramides based ingredient cream and the placebo product [19] (Table 3, Figure 5).

**Table 3.** Mean values, standard deviations, variations, and statistical significance (*t*-test) of transepidermal water loss on 20 analyzed subjects.

| Products | T0 | T4 Weeks | Variation (%) T4 Weeks-T0 | P-Level T0 vs. T4 Weeks |
|---|---|---|---|---|
| ceramides and trehalose ingredient based cream | mean 9.96 st. dev. 2.43 | mean 7.32 st. dev. 1.19 | −2.64 (−26.5%) | *p* < 0.0001 |
| placebo cream | mean 10.35 st. dev. 2.56 | mean 9.12 st. dev. 2.00 | −1.23 (−11.9%) | *p* < 0.05 |

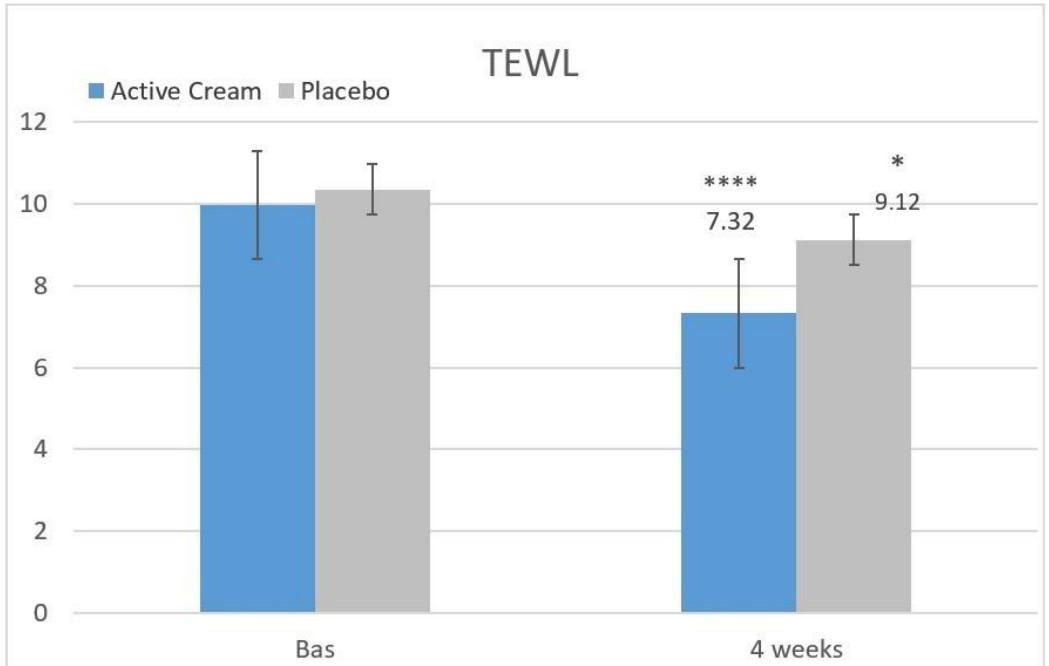

**Figure 5.** Mean values of transepidermal water loss recorded for cream containing trehalose and ceramides based ingredient and placebo product. TEWL values (+/− SD) show a reduction of 26.5% for cream containing 2% trehalose and ceramides based ingredient site after 4 weeks (**** *p* < 0.0001 (*t*-test); * *p* < 0.05 (*t*-test); n = 20 subjects; Bas = Baseline, T0; 4 weeks = after 4 weeks of application).

## 4. Results

A statistical analysis has been performed for the instrumental evaluation between the nontreated area (T0) and the trehalose and ceramides based in cream treated area (TF) and then compared to placebo area for both parameters (hydration and TEWL).

Evaluation of:

(1) T0 vs T4 weeks in the area treated with formula containing trehalose and ceramides based ingredient.

(2) T0 vs T4 weeks in the placebo-treated area.

(3) The variations (TF−T0) obtained in the active area have been statistically compared with variations (TF−T0) obtained in the placebo area.

A statistically significant difference between active and placebo product was detected after 4 weeks of treatment.

Variation % vs placebo (var cream containing ingredient−var placebo/var placebo × 100)

(1) Hydration (18.7 − 15.4)/15.4 × 100 = +21.4%

(2) TEWL (2.64 − 1.23)/1.23 × 100 = −114.6%

The instrumental evaluations performed at the beginning and after 4 weeks of treatment with the formula based on trehalose and ceramides based ingredient gave the following results:

(1) A statistically significant increase in skin hydration (+69.5%);

(2) A statistically significant improvement in transepidermal water loss (−26.5%).

The efficacy of the formulation containing the ingredient object of this study was confirmed by a statistically significant difference between cream containing ceramides and trehalose based ingredient and placebo product for both the considered parameters [19].

## 5. Conclusions

According to the in vivo efficacy tests, the object of this study is a new ingredient composed of ceramides, trehalose, lecithin, and cholesterol, able to restore the physiological hydro-lipidic film and the protective skin barrier structure, at the same time ensuring a healthy level of hydration by reducing the TEWL (transepidermal water loss). Its film-forming properties result in a soothing and protective action against itching, burning, and aggressions by external agents and make it the ideal compound for atopic dermatitis symptoms alleviation.

**Author Contributions:** Writing—review and editing, L.G. and S.U.; original draft preparation, L.G.; investigations, S.U.; data curation, S.U.; methodology, L.R.; formal analysis, L.R.; project administration, M.F.; validation, E.B.; supervision, E.B.; conceptualization, E.B.; reviewing and validation, N.C.

**Funding:** This research was funded by Sinerga SPA, VAT NUMBER 12950420153.

**Acknowledgments:** Authors sincerely acknowledge Abich Elena Bocchietto for TEM analysis and Adriana Bonfiglio from ISPE for the clinical assessment.

**Conflicts of Interest:** The authors declare no conflict of interest.

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
