# Peer review of "Evaluation of the Filming and Protective Properties of a New Trehalose and Ceramides Based Ingredient"

_cosmetics, doi:10.3390/cosmetics6040062_

Round 1
Reviewer 1 Report
The paper is interesting and show something new for modern formulations of topic treatments
Author Response
Dear reviewer,
Thanks a lot for your feedback.
I’m glad to hear that the paper sounds interesting to you.
Kind Regards,
Letteria Greco
Reviewer 2 Report
Comments to the authors,
Some of the required corrections and amendments that I asked in the first review are not considered. For example, why there are two conclusions for the paper and "in vivo" should be in italic font throughout the article. Also, it is suggested that a native English speaker edits the manuscript for a more coherent and confluent text.
Author Response
Dear reviewer,
Thanks a lot for your comments.
I just revised the article and put in italic all the “in vivo”. I also put only a final conclusion, for in vivo I added the title “Results”.
I hope titles and subtitles are more clear. I also asked to a native USA English speaker to revise the article, it should be more fluent and coherent.
Kind regards,
Letteria Greco

This manuscript is a resubmission of an earlier submission. The following is a list of the peer review reports and author responses from that submission.
Round 1
Reviewer 1 Report
The Research question and aim is relevant, however this paper have serious methodological issues and Research has not been conducted according to good research standards.
Major ethical issues:
1) Conflicts of interests have not been declared, even when it seems that the Authors seem to work for the cosmetic industry.
Some of the seriuos methodological issues:
Double blinding is not properly described.
No information about sampling.
Lack of a Control Group.
Lack of background information of the intervention Group.
Results for skin barrier measurements are not properly presented.
Strategy for statitical analysis is wrong.
Lack of discussion of the results.
The conclusion is not supported by the study design.
Author Response
Dear reviewer,
Thank you so much for your kind feedback.
Please, see above my replies with edits I did to the article following your comments.
Major ethical issues:
- Conflicts of interests have not been declared, even when it seems that the Authors seem to work for the cosmetic industry. à we now declare that three authors work for Sinerga Group and Sinerga SPA.
Some of the seriuos methodological issues:
- Double blinding is not properly described. No information about sampling. Lack of a Control Group. Lack of background information of the intervention Group. à In order to clarify, we added more details for the first and the second test.
- Lack of discussion of the results. The conclusion is not supported by the study design.—> I reworked the results, trying to structure better tables and graphs.
I hope to have informed you well.
Kind regards,
Letteria Greco

Reviewer 2 Report
Comments for authors,
In the manuscript entitled “Evaluation of the filming and protective properties of a new Trehalose and Ceramides active blend”, authors describe a new active blend of trehalose and ceramides delivered through the lipophilic vesicles for skin disorders that patients have skin barrier dysfunction.
It is suggested to extensively reorganize the paper sections, and it would be much better if the authors make more clarification. The introduction is almost fine; however, the other sections should be revised because it is confusing. There is a short and incomplete “Materials and Methods” on page 3. There is no information on the patients, the IRB of the study, and the selection criteria, only on page 6, there are some explanations on the patients. Then on page 4, line 122, there is a “Results and Discussion” that without any explanation started with statistical analysis of Evaporiometry.
Also, on page 5, there is an irrelevant explanation of Dermatitis and its different types that should be transferred to the introduction part of the manuscript.
Again, on page 7, there is a “Results” topic that presented the results for “Skin Hydration” and “Trans-Epidermal Water Loss (TEWL).” The statistical analysis should be more clarified since it seems that there is a bias in the data presented. In conclusion, the percentages are in the patients’ group, and they are not presented as compared to the Placebo group. Therefore, the following statements in page 8, are not correct:
- a statistically significant increase in the skin hydration (+69.5%);
- a statistically significant improvement in the trans-epidermal water loss (-26.5%).
Moreover, there are multiple typos and English mistakes in the manuscript. It is suggested to request a native English speaker to revise the paper. For example:
“in vivo” should be in italic;
On page 3, line 109 it is Materials, not Matherials;
The title of all figures should be uniform as Figure 1, Figure 2, etc. not Graph 2.
On page 4, line 148, did you refer to Figures 2 and 3? What are Graph 1 and graph2?
On page 4, line 124 it should be “…. shows a very good trend…”
On page 4, line 145 it is trehalose not threhalose
Author Response
1- Dear reviewer,
Thank you so much for your kind feedback.
Please, see above my replies with edits I did to the article following your comments.
Comments for authors,
In the manuscript entitled “Evaluation of the filming and protective properties of a new Trehalose and Ceramides active blend”, authors describe a new active blend of trehalose and ceramides delivered through the lipophilic vesicles for skin disorders that patients have skin barrier dysfunction.
It is suggested to extensively reorganize the paper sections, and it would be much better if the authors make more clarification. The introduction is almost fine; however, the other sections should be revised because it is confusing.
- There is a short and incomplete “Materials and Methods” on page 3. There is no information on the patients, the IRB of the study, and the selection criteria, only on page 6, there are some explanations on the patients. à We improved the first description of patients info and exclusion criteria on page 3
- Then on page 4, line 122, there is a “Results and Discussion” that without any explanation started with statistical analysis of Evaporimetry. à I added explanation about Corneometry and Evaporimetry.
- Also, on page 5, there is an irrelevant explanation of Dermatitis and its different types that should be transferred to the introduction part of the manuscript. à I put the description of Dermatitis right before the test, after Introduction
- Again, on page 7, there is a “Results” topic that presented the results for “Skin Hydration” and “Trans-Epidermal Water Loss (TEWL).” The statistical analysis should be more clarified since it seems that there is a bias in the data presented. In conclusion, the percentages are in the patients’ group, and they are not presented as compared to the Placebo group. Therefore, the following statements in page 8, are not correct:
· a statistically significant increase in the skin hydration (+69.5%);
· a statistically significant improvement in the trans-epidermal water loss (-26.5%).
à Regarding this, statistical analysis has been performed for instrumental evaluation between non treated area and active cream treated area and then compared to placebo group for both parameters (hydration and TEWL).
T0 vs T4 weeks in the treated area has been evaluated.
T0 vs T4 weeks in no treated area has been evaluated.
This values have been then compared with placebo group.
Variation % vs placebo (var active cream–var placebo/var placebo x 100)
-Hydration (18.7-15.4)/15.4 x 100 = 21.4%
- TEWL (2.64-1.23)/1.23 x 100 = -114.6%
The variation, expressed as mean value and as %, is statistically significant.
- Moreover, there are multiple typos and English mistakes in the manuscript. It is suggested to request a native English speaker to revise the paper à I asked a revision to a native speaker.
For example:
“in vivo” should be in italic; à corrected
On page 3, line 109 it is Materials, not Matherials; à corrected
The title of all figures should be uniform as Figure 1, Figure 2, etc. not Graph 2. à corrected
On page 4, line 148, did you refer to Figures 2 and 3? What are Graph 1 and graph2? à corrected
On page 4, line 124 it should be “…. shows a very good trend…” à corrected
On page 4, line 145 it is trehalose not threhalose à corrected
I hope to have informed you well.
Kind regards,
Letteria Greco

Reviewer 3 Report
the paper is innovative and interesting for further developments in treatment of neglected skin diseases
Author Response
Dear reviewer, thanks a lot for your encouraging feedback.
Please, see attached updated version.
Kind regards,
Letteria Greco

Round 2
Reviewer 1 Report
Whilst some modificationss have been made, research has not been performed properly.
Author Response
Dear reviewer,
to clarify statistical analysis, I had a meeting with the person who performed it.
That’s what I added in the conclusion of instrumental evaluation:
Statistical analysis has been performed for instrumental evaluation between non treated area (T0) and active cream treated area (TF) and then compared to placebo area for both parameters (hydration and TEWL).
T0 vs T4 weeks in the Active treated area has been evaluated.
T0 vs T4 weeks in the Placebo treated area has been evaluated.
The variations (TF-T0) obtained in the active area have been statistically compared with variations (TF-T0) obtained in the placebo area.
Variation % vs placebo (var active cream–var placebo/var placebo x 100)
-Hydration (18.7-15.4)/15.4 x 100 = 21.4%
- TEWL (2.64-1.23)/1.23 x 100 = -114.6%
The variation, expressed as mean value and as %, is statistically significant
I hope this will make the conclusion and its relevance more clear.
Thanks and regards,
Letteria Greco

Reviewer 2 Report
Thank you for your amendments and corrections. There are still some minor points that need to be addressed.
It is also suggested to highlight in bold font the headings of “Introduction, Materials and Methods, Results and discussion and conclusion” and if you want to emphasize on subtitles of each section use a different font or size. You have two conclusion sections on page 9; please correct accordingly. Also, please remove the repeated “Results” title on page 8, line238 and the numbers from the headings. You provide a good explanation in your response letter for the statistical analysis method and add a good clarification on Figures 2 and 3. Please pay attention that for TEWL the numbers are negative and the variation would be (-2.64-1.23)/-1.23 x 100 = 314.6% but not (2.64-1.23)/1.23 x 100 = -114.6%. Please add the following explanations from your response to the “Results and Discussion” section.
T0 vs. T4 weeks in the treated and untreated area have been evaluated. These values have been then compared with the placebo group.
Variation % vs placebo (variation in active cream group–variation in placebo group/variation in placebo group x 100)
- Hydration (18.7-15.4)/15.4 x 100 = 21.4%
- TEWL (-2.64-1.23)/-1.23 x 100 = -314.6%
The variation, expressed as mean value and as %, is statistically significant.
In addition, please correct “Bas” to “Baseline” in Figures 2 and 3 and “wks” in Figures 4 and 5 to “weeks”.
Author Response
Dear reviewer,
Regarding titles and subtitles, I followed your advice and put in bold only relevant titles. To highlight subtitles I put them in a different font size. I hope this is more clear now. I also checked the two conclusions and results.
to clarify statistical analysis, I had a meeting with the person who performed it.
That’s what I added in the conclusion of instrumental evaluation:
Statistical analysis has been performed for instrumental evaluation between non treated area (T0) and active cream treated area (TF) and then compared to placebo area for both parameters (hydration and TEWL).
T0 vs T4 weeks in the Active treated area has been evaluated.
T0 vs T4 weeks in the Placebo treated area has been evaluated.
The variations (TF-T0) obtained in the active area have been statistically compared with variations (TF-T0) obtained in the placebo area.
Variation % vs placebo (var active cream–var placebo/var placebo x 100)
-Hydration (18.7-15.4)/15.4 x 100 = 21.4%
- TEWL (2.64-1.23)/1.23 x 100 = -114.6%
The variation, expressed as mean value and as %, is statistically significant
I hope this will make the conclusion and its relevance more clear.
I also modified graphs putting Baseline instead then bas and also “weeks”.
Thanks and regards,
Letteria Greco
